# Introducing synthetic thermostable RNase inhibitors to single-cell RNA-seq

Joyce Carol Noble [1], Antonio Lentini [1], Michael Hagemann-Jensen [2], Rickard Sandberg [2] & Björn Reinius [1] ✉

Single-cell RNA-sequencing (scRNAseq) is revolutionizing biomedicine, propelled by advances in methodology, ease of use, and cost reduction of library preparation. Over the past decade, there have been remarkable technical improvements in most aspects of single-cell transcriptomics. Yet, little to no progress has been made in advancing RNase inhibition despite maintained RNA integrity being critical during cell collection, storage, and cDNA library generation. Here, we demonstrate that a synthetic thermostable RNase inhibitor (SEQURNA) yields single-cell libraries of equal or superior quality compared to ubiquitously used protein-based recombinant RNase inhibitors (RRIs). Importantly, the synthetic RNase inhibitor provides additional unique improvements in reproducibility and throughput, enables new experimental workflows including retained RNase inhibition throughout heat cycles, and can reduce the need for dry-ice transports. In summary, replacing RRIs represents a substantial advancement in the field of single-cell transcriptomics.

The capture of intact RNA is required for RNAseq methods to accurately record the transcriptome of the analyzed sample material. scRNAseq is particularly sensitive to RNA degradation and losses due to the minuscule copy numbers of individual transcripts present in the cell[1,2]. Thus, the inclusion of RRIs, which are in vitro synthesized RNase-binding proteins, is nearly universal during cell capture, storage, cell lysis, and reverse transcription (RT) in scRNAseq protocols. However, the use of RRIs is inconvenient not only due to its relatively high cost fraction of the scRNAseq libraries but also due to RRI degradability, which can introduce batch variation in library yield and quality with production lot, storage time, and temperature exposures of the inhibitor. Furthermore, thermal sensitivity of RRIs imposes restrictions on the temperature ranges permissible in experimental protocols and laboratory procedures, while also complicating the logistics of exchanging reagents and biological samples. Due to their unmatched performance, RRIs continue to be the leading standard in scRNAseq, and replacing them has so far proven non-trivial. In certain scRNAseq protocols, developers have substituted RRIs with alternatives like guanidinium salts, beta-mercaptoethanol, or diethyl pyrocarbonate[3,4]. While these chemicals decrease RNA degradation, their performance in scRNAseq is not on par with RRIs. They are furthermore toxic and carry characteristics that are counterproductive in subsequent steps of scRNAseq library preparation, including their chaotropic nature, reduction capability, or susceptibility to temperature and buffer variations. These properties adversely impact RT and PCR processes, ultimately compromising the quality of the sequencing library. To offer a full substitute for RRIs without sacrificing library quality, we developed SEQURNA, a synthetic thermostable RNase inhibitor which consists of a mix of non-toxic organic molecules that furthermore do not depend on any toxic reducing agents that are needed for conventional RRI functionality (e.g., DTT or beta-mercaptoethanol).

This work introduces, for the first time, the successful substitution of RRIs with a synthetic thermostable RNase inhibitor across various scRNAseq applications, leading to maintained or even enhanced sequencing library quality. The adoption of the synthetic RNase inhibitor paves the way for novel experimental workflows and logistical improvements, poised to revolutionize single-cell genomics methodologies ahead.

[1]Department of Medical Biochemistry and Biophysics, Karolinska Institutet, Stockholm, Sweden. [2]Department of Cell and Molecular Biology, Karolinska Institutet, Stockholm, Sweden. ✉e-mail: bjorn.reinius@ki.se

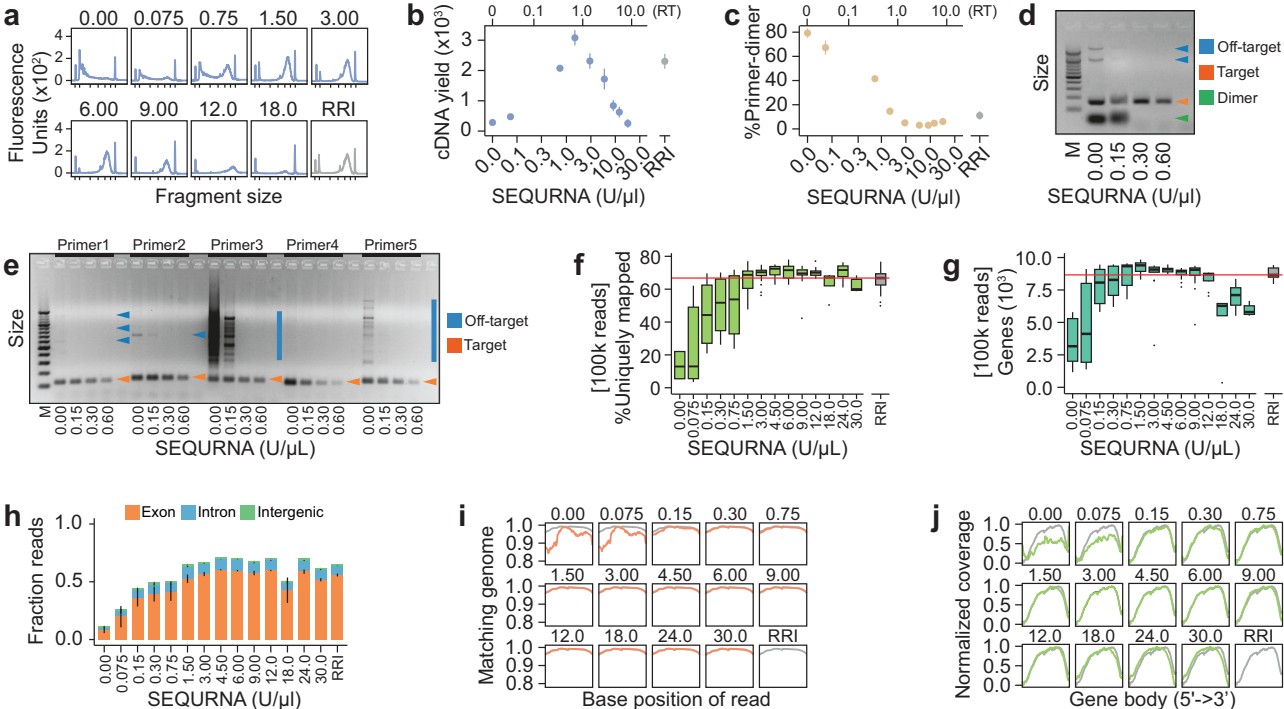

**Fig. 1 | Replacing RRIs with a synthetic thermostable RNase inhibitor in mini-bulk RNA-seq library generation. a** Bioanalyzer traces of Smart-seq2 (SS2) cDNA libraries generated from 100 pg mouse RNA using varying SEQURNA concentrations (0–18 U/μl) or standard recombinant RNase inhibitor (RRI) in lysis buffer. Tick marks on the x-axis correspond to 35, 100, 300, 500, 1000, 3000, and 10380 base pairs. **b** Quantified cDNA yield of bulk SS2 cDNA libraries (fragment size 200–10,000 bp). SEQURNA concentration in lysis buffer on the lower x-axis and resulting concentration in RT on the upper x-axis. Data shown as mean (dot) and standard error (line). $n = 2$–8 traces per condition (54 traces analyzed). **c** Percent primer-dimer in SS2 bulk cDNA libraries (fragment size 20–50 bp) for the samples in (**b**). **d** Gel electrophoresis image of products formed using an eGFP plasmid as template and varying the concentration of SEQURNA during PCR. Off-target, target and dimer bands indicated by blue, orange and green arrows, respectively. M = marker ladder. **e** Gel electrophoresis image of products formed using human gDNA as template and five primer sets targeting *TUBA1A*. Off-target and target bands indicated in blue and orange, respectively. M = marker ladder. Box plots of percent uniquely mapped sequencing reads to the reference genome (**f**) and number of genes detected (**g**) for SS2 libraries generated from 100 pg mouse RNA. Data shown as median, interquartile range (IQR) and 1.5x IQR for n = 5–35 replicates per condition (total = 195). **h** Stacked bar plot of fraction reads mapping to exonic (orange), intronic (blue), or intergenic (green) regions of the mouse genome for bulk SS2 libraries. Data shown as mean with lines indicating standard error. **i** Line plot of fraction of bases along sequencing reads matching the reference transcripts, indicating read quality, for bulk SS2 libraries. **j** Line plot of normalized coverage along transcripts for bulk SS2 libraries. Data underlying relevant plots is provided in Source Data and P-values for statistical comparisons are available in Supplementary Data 1.

## Results

### A synthetic thermostable RNase inhibitor for scRNAseq

To assess its utility in scRNAseq library preparation, we started by systematically testing concentration ranges of SEQURNA in the Smart-seq2 (SS2) protocol[5], which provides full-length read coverage across transcripts enabling evaluation of polymerase processivity. The standard SS2 protocol includes a cell lysis and RNA denaturation step which involves heating the sample to 72 °C before RT, requiring the addition of fresh RRI in the RT mix due to thermosensitivity of the protein-based RNase inhibitor. Conversely, the synthetic RNase inhibitor was added only in the lysis buffer, retaining effectiveness throughout the denaturation step and eliminating the need for its addition in the RT mix. We experimented with lysis buffers containing a range of SEQURNA (0–30 U/μl; resulting in 0–13.5 U/μl in RT) and 100 pg purified mouse RNA, performed SS2 library amplification, and evaluated the resulting cDNA by capillary electrophoresis. Within a defined concentration range of SEQURNA (1.5–6 U/μl) the cDNA fragment size distribution was on par with the standard SS2 protocol with RRIs, whereas the cDNA yield was similar or increased with SEQURNA (Fig. 1a, b). At lower and higher SEQURNA concentrations, the cDNA yield declined due to RNA degradation and RT inhibition, respectively. Intriguingly, we noticed that SEQURNA also slightly reduced the fraction of primer-dimer content in the cDNA libraries at high concentrations (Fig. 1a, c). To better explore how SEQURNA affects PCR specificity, we performed a previously established PCR-based dimer assay using primers intentionally designed to self-hybridize[6] and an eGFP plasmid as template (Methods). At increasing SEQURNA concentration, primer-dimers were indeed eliminated whereas the target amplicon was preserved (Fig. 1d). We also designed *TUBA1A*-targeting primer pairs of varying tendencies to form unspecific products and found that SEQURNA could prevent unintended bands and smear products amplified from human gDNA (Fig. 1e). By melting-curve analysis, we found that the increased PCR stringency was not the mere result of altering melting temperature (Supplementary Fig. 1a). Notably, enhanced stringency may increase the proportion of informative fragments in scRNAseq libraries (Fig. 1c). To examine the information yield and data quality of the SS2 libraries, we sequenced libraries generated from mouse RNA in presence of SEQURNA or standard RRI, and downsampled the resulting reads to allow even comparison (Methods). We analyzed library quality metrics in terms of genomic and transcriptomic mappability, base substitution rates, insertion and deletions (indels), likelihood of mismatches within the read, and average indel length. SEQURNA conditions 1.5–12 U/μl in the lysis buffer (0.675–5.4 U/μl in RT) yielded libraries of similar quality as RRI with respect to all parameters investigated, including the number of detected genes, fraction reads mapped to exons, base-along-read likelihood of matching genome, and gene body coverage (Fig. 1f–j and Supplementary Fig. 1b, c). Additionally, substitution and indel rates

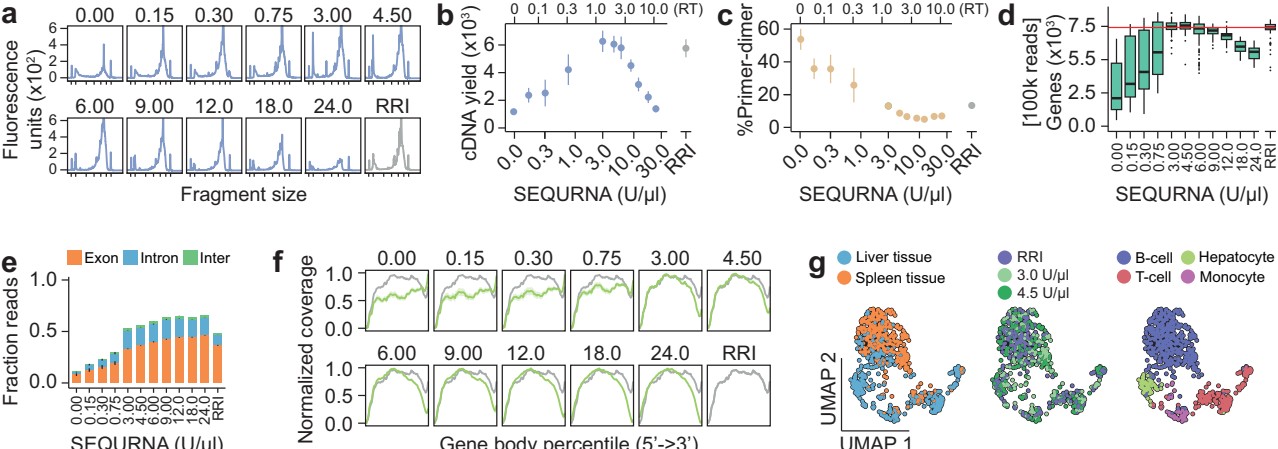

**Fig. 2 | Performance of SEQURNA in single-cell Smart-seq2. a** Bioanalyzer traces of Smart-seq2 (SS2) cDNA libraries from HEK293FT single cells using varying SEQURNA concentrations (0–24 U/μl) or standard recombinant RNase inhibitor (RRI) in lysis buffer. Tick marks on the x-axis correspond to 35, 100, 300, 500, 1000, 3000, and 10380 base pairs. **b** Quantified cDNA yield of single-cell SS2 cDNA libraries from HEK293FT cells (fragment size 200–10,000 bp). SEQURNA concentration in lysis buffer on the lower x-axis and resulting concentration in RT on the upper x-axis. Data shown as mean (dot) and standard error (line). $n = 3$–6 traces per condition (50 traces analyzed). **c** Percentage primer-dimer of single-cell SS2 cDNA libraries from HEK293FT cells (fragment size 20–50 bp) for the samples in (**b**). **d** Box plot of number of genes detected in HEK293FT single-cell SS2 libraries. Data shown as median, interquartile range (IQR), and 1.5x IQR. $n = 37$–94 per condition (704 libraries analyzed). **e** Stacked bar plot of fraction reads mapping to exonic (orange), intronic (blue), or intergenic (green) regions of the human genome for single-cell HEK293FT SS2 libraries. Data shown as mean with lines indicating standard error, $n = 37$–94 per condition. **f** Line plot of normalized coverage along transcripts for single-cell HEK293FT SS2 libraries, $n = 37$–94 per condition. **g** Uniform manifold approximation and projection (UMAP) for single-cell SS2 libraries in mouse cells derived from liver ($n = 349$ cells) and spleen ($n = 368$ cells) tissue. Cell sorting was gated for small cell size to enrich cells with low RNA content (Supplementary Fig. 5). Colors indicate tissue origin from liver (blue) or spleen (orange) (left), SEQURNA concentrations 3.0 U/μl (light green), 4.5 U/μl (dark green) or RRI (purple) (middle) and cell type identified as B-cell (blue), T-cell (red), hepatocyte (green) or monocyte (purple) (right). Data underlying relevant plots is provided in Source Data and $P$-values for statistical comparisons are available in Supplementary Data 1.

were on par between SEQURNA and RRI in the relevant concentration range (Supplementary Fig. 1d–i), consistent with unaffected polymerase fidelity.

Together, these results demonstrate a successful working range (-1 order of magnitude) of SEQURNA in SS2 library generation and that the cDNA yield curve correlates with downstream quality metrics. Therefore, the notion of a defined working range of the synthetic RNase inhibitor is a key consideration, to be identified for individual scRNAseq protocols as they differ in buffer composition, volumetric changes between cell lysis and RT reaction, and reverse transcriptase used.

### Efficacy in stress conditions and RNase-rich samples

A synthetic and thermostable RNase inhibitor might better tolerate variable storage conditions, thus we explored the efficacy of SEQURNA in SS2 library generation after first subjecting inhibitor stocks to various "harsh" treatments (Methods) prior to use in lysis buffers, such as heating it to 37 °C or 50 °C for 24 h, freeze-thawing up to six times, vortexing for 24 h, and drastically altering pH (pH 4-10), remarkably observing intact cDNA traces in each treatment (Supplementary Fig. 2). Importantly, the robustness of the synthetic inhibitor enables a wide array of experimental treatments and conditions, expanding the range of potential applications and introducing multiple new experimental pathways in single-cell omics techniques.

Encouraged by the results on purified RNA, we FACS sorted individual cultured human embryonic kidney cells (HEK293FT) into 96-well plates containing lysis buffer and 0–24 U/μl SEQURNA, or standard SS2 buffer with RRI (Methods), and analyzed the resulting single-cell SS2 libraries. At optimal SEQURNA concentration (peaking around 2–3 U/μl in lysis buffer) (Fig. 2a–c), the sequencing libraries and generated transcriptome data were on par with standard SS2 libraries generated using RRI (Fig. 2d–f and Supplementary Fig. 3), with similar capture of biological signals, exemplified by the cell-cycle (Supplementary Fig. 4). Together, these results demonstrate successful

capture of minute biological signals in scRNAseq data generated with a synthetic RNase inhibitor, on par with RRIs.

We then challenged the synthetic RNase inhibitor by isolating small cells (low RNA content) dissociated from mouse spleen and liver (Supplementary Fig. 5a, b)—tissues known to be of high and moderate RNase content, respectively. Single cells were sorted into lysis buffer containing either RRI or SEQURNA (3 or 4.5 U/μl). The resulting library quality of the three conditions were highly similar (Supplementary Fig. 5c–p). We visualized the relationship between cells and tissue origin by Uniform Manifold Approximation and Projection (UMAP) and observed that the distribution of SEQURNA- and RRI-treated samples was well represented throughout the projection and indicated little or no difference in gene and cell-type detection between the conditions (Fig. 2g). In brief, independent of inhibitor type, we identified four distinct clusters representing B-cells, T-cells, monocytes/macrophages (clusters 1,2,4) as well as hepatocytes (cluster 3) (Supplementary Fig. 6), in line with liver and spleen lymphocyte/monocyte populations identified in previous studies[7,8], demonstrating successful transcriptome capture in small cells from RNase-rich tissues.

### Performance in UMI-based and commercial protocols

To demonstrate broad utility of the synthetic RNase inhibitor in next-generation sequencing protocols, we applied SEQURNA to Smart-seq3 (SS3)[9] on purified mouse RNA and single HEK293FT cells. At identified optimal concentration (0.06–0.6 U/μl SEQURNA in the lysis buffer; 0.03–0.3 U/μl during RT) it produced maximum cDNA yield (Fig. 3a–d) and sequencing data of parallel or superior quality metrics to standard SS3 using RRI (Fig. 3e–g and Supplementary Fig. 7), including similar UMI counts (Fig. 3h). Using SEQURNA in one commercial bulk RNAseq kit (TruSeq RNA Library Preparation Kit v2) and two commercial single-cell RNAseq kits (NEBNext Single Cell/Low Input RNA Library Prep Kit, and QIAseq FX Single Cell RNA Library Kit) demonstrated SEQURNA libraries of high quality (Supplementary Fig. 8). This implicates broad

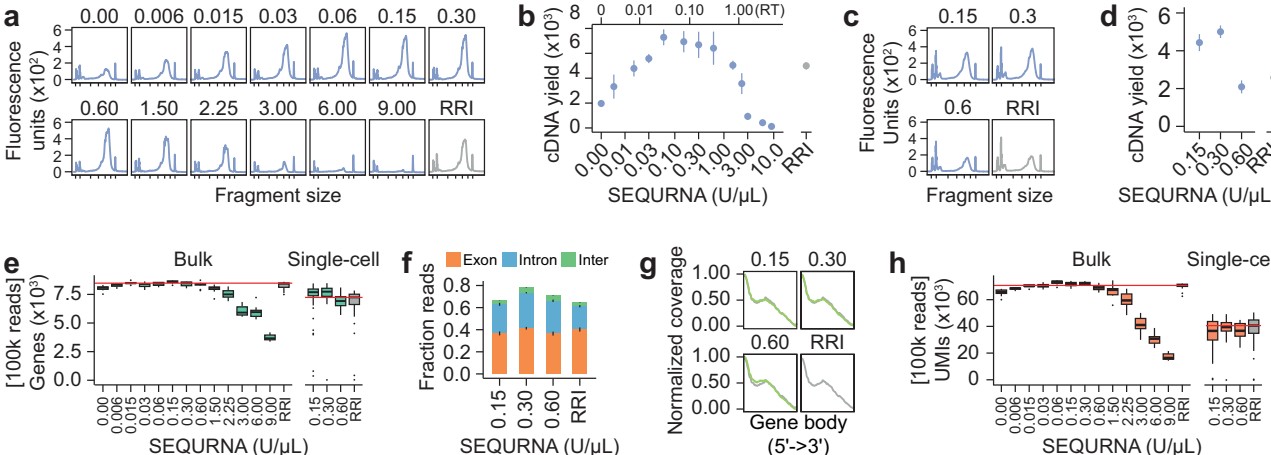

**Fig. 3 | Performance of SEQURNA in mini-bulk and single-cell Smart-seq3.**
**a** Bioanalyzer traces of Smart-seq3 (SS3) cDNA libraries generated from 100 pg mouse RNA (bulk) using varying SEQURNA concentration (0–9 U/μl) or standard SS3 recombinant RNase inhibitor (RRI) in lysis buffer. Tick marks on the x-axis correspond to 35, 100, 300, 500, 1000, 3000, and 10380 base pairs. **b** Quantified cDNA yield of bulk SS3 cDNA libraries (fragment size 200–10,000 bp). Data shown as mean (dot) and standard error (line). $n = 3$–5 traces per condition (51 traces analyzed). **c** Bioanalyzer traces of single-cell SS3 cDNA libraries from HEK293FT cells using varying SEQURNA concentrations (0.15, 0.3, 0.6 U/μL) or standard SS3 RRI in lysis buffer. **d** Quantified cDNA yield of single-cell SS3 cDNA libraries from HEK293FT cells (fragment size 200–10,000 bp). Data shown as mean (dot) and standard error (line). $n = 11$–14 traces per condition (53 traces analyzed). **e** Box plot of number of genes detected in SS3 libraries for 100 pg mouse bulk RNA

(left, $n = 7$–16 libraries per condition, total = 170) and single-cell HEK293FT (right, $n = 31$–45 cells per condition, total = 158). Data shown as median, interquartile range (IQR), and 1.5x IQR. **f** Stacked bar plot of fraction reads mapping to exonic (orange), intronic (blue), or intergenic (green) regions of the human genome for single-cell HEK293FT SS3 libraries. Data shown as mean with lines indicating standard error. $n = 31$–45 cells per condition. **g** Line plot of normalized coverage along transcripts for single-cell HEK293FT SS3 libraries, with characteristic bias towards the 5'-UMI-containing end. $n = 31$–45 cells per condition. **h** Box plot of number of unique molecular identifiers (UMIs) detected in SS3 libraries for 100 pg mouse bulk RNA (left, $n = 7$–16 samples per condition) and single-cell HEK293FT ($n = 31$–45 cells per condition). Data shown as median, interquartile range (IQR), and 1.5x IQR. Data underlying relevant plots is provided in Source Data and $P$-values for statistical comparisons are available in Supplementary Data 1.

applicability of the synthetic RNase inhibitor across diverse experimental conditions and RNAseq methods.

**Improved RNA stability and workflow flexibility**

A thermostable RNase inhibitor might offer unique RNA integrity protection during sample handling, from the collection of cells until stable cDNA is produced. Based on our results, we hypothesized that cells collected in SEQURNA lysis buffer would be more tolerant to extended waiting times compared to conventional RRIs, a feature of importance for large-scale scRNAseq projects. To this end, cells sorted into SS2 lysis buffer and 0–12 U/μl SEQURNA were stored at ambient temperature (25 °C), cold (4 °C) or ultra-low temperature (−80 °C) for 1, 4, 7, or 14 days before SS2 libraries were generated. Expectedly, we observed that cells stored in lysis buffer lacking inhibitor resulted in degraded cDNA traces (Supplementary Fig. 9a–d). Intriguingly, cDNA profiles obtained from SEQURNA samples even at 4 days of storage at room temperature (25 °C) showed lower amounts of degradation (Supplementary Fig. 9c) with workable RNA integrity even up to day 4, and remarkably, throughout the full 14-day experiment when stored refrigerated (4 °C) (Supplementary Fig. 9d). Although this observation based on cDNA traces alone should not be interpreted as a complete lack of RNA degradation over storage time, it does indicate a surprising stability of cellular RNA over extended time when kept in lysis buffer containing synthetic RNase inhibitor.

To systematically characterize the capacity of SEQURNA to extend quality in longer-term stored RNA from single cells, we conducted a large-scale sequencing experiment utilizing the newly developed Smart-seq3xpress protocol (SS3xpress)[10]. This protocol is a miniaturized adaptation of SS3 which not only utilizes a markedly lowered reaction volume that drastically reduces cost but also streamlines the library-preparation process by completely circumventing the cDNA purification step before tagmentation, instead proceeding with tagmentation and indexing PCR immediately after a cDNA dilution step. This necessitates compatibility between transposase Tn5 and

SEQURNA in the relevant concentration range. Thus, we experimentally characterized how SEQURNA affected Tn5 tagmentation of double-stranded cDNA. Surprisingly, we observed that SEQURNA was not only compatible with Tn5 activity but that it increased product yield when SEQURNA was added in low concentration (0.0025–0.1 U/μl) (Fig. 4a, b) and inhibition was only observed above 0.25 U/μl. While the exact mechanism behind this interaction remains to be determined, this effect might be beneficial in single-cell multi-omics experiments utilizing Tn5.

Next, we sorted HEK293FT cells into 384-well plates with SS3xpress lysis buffer containing SEQURNA (0, 0.06, 0.15, 0.3, 0.6, 1.5, or 3.0 U/μl) or standard SS3xpress lysis buffer with RRI. To avoid potential inter-plate batch effects, all inhibitor conditions were represented within each plate. Two plates were kept at −80 °C immediately after cell sorting (representing the day 0 time point) while the other plates were stored at either room temperature (25 °C) or refrigerated (4 °C) for up to 14 days, and thereafter processed into SS3xpress libraries and sequenced (Methods). In the absence of cDNA traces (due to direct tagmentation in SS3xpress), we identified the optimal concentration range of SEQURNA directly from read mapping statistics from day 0, where the optimal range was 0.06–0.3 U/μl SEQURNA in the lysis buffer (resulting in 0.045-0.225 U/μl in RT reaction) (Fig. 4c–e), i.e., similar to that identified for SS3. Interestingly, we observed that SEQURNA in the optimal range (0.06–0.3 U/μl) consistently produced improved libraries (more genes and UMIs detected per cell) compared to standard SS3xpress using RRI, indicating that SEQURNA can enhance the established SS3xpress protocol. Next, we analyzed downsampled data from cells stored at 25 °C and 4 °C, only considering the working range of SEQURNA concentrations (0.06–0.6 U/μl), no inhibitor (0 U/μl), and standard RRI. As expected, we observed the overall trend of decreasing number of genes and UMIs detected per cell over time in storage, and a more rapid degradation at 25 °C compared to 4 °C (Fig. 4f–i). However, degradation progressed markedly slower for cells in SEQURNA lysis buffer, an effect

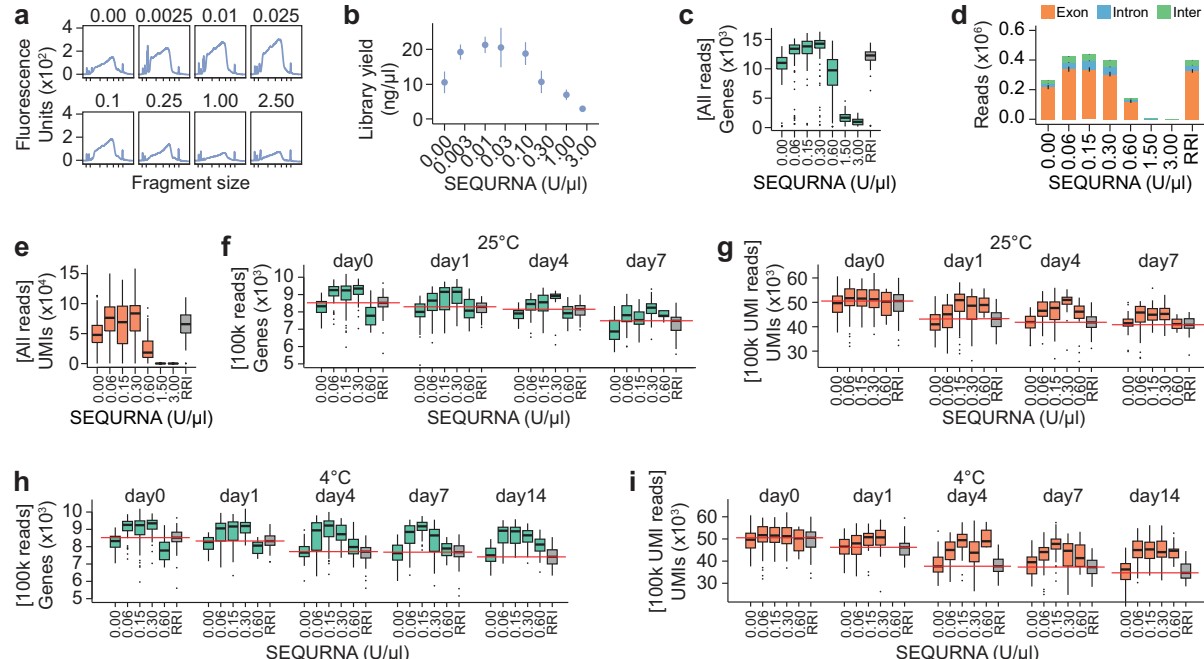

**Fig. 4 | Characteristics of Smart-seq3xpress single-cell libraries after long-term storage in SEQURNA lysis buffer. a** Bioanalyzer traces of Tn5-tagmented and amplified libraries of HEK293FT single-cell cDNA varying SEQURNA concentration in the tagmentation reaction. X-axis ticks correspond to 35, 100, 300, 500, 1000, 3000, and 10380 base pairs. **b** Quantified library yield of libraries in (**a**). Data shown as mean ± standard error of $n = 9$ per condition (72 total). **c** Box plot of number of genes detected in Smart-seq3xpress (SS3xpress) libraries of HEK293FT cells collected in SS3xpress lysis buffer containing no inhibitor (0 U/μl), SEQURNA (0.06, 0.15, 0.3, 0.6, 1.5, or 3.0 U/μL), or standard SS3xpress recombinant RNase inhibitor (RRI). $n = 88–96$ cells per condition, total = 757 (Day0 only). **d** Stacked bar plot of reads mapping to exonic (orange), intronic (blue), or intergenic (green) regions for single-cell HEK293FT SS3xpress corresponding to cells in (**c**). Data shown as mean ± standard error. **e** Box plots of number of unique molecular identifiers (UMIs) corresponding to cells in (**c**). **f** Box plot of number of genes detected in

single-cell HEK293FT SS3xpress varying RNase inhibitor conditions in the lysis buffer and storing cells for 0, 1, 4, or 7 days at 25 °C. $n = 57–93$ cells per condition for Day 0 and $n = 6–48$ cells per condition for Days 1–7, total = 1220. **g** Box plot of number of UMIs detected corresponding to cells in (**f**). **h** Box plot of number of genes detected in single-cell HEK293FT SS3xpress varying RNase inhibitor conditions in the lysis buffer and storing cells for 0, 1, 4, 7, or 14 days at 4 °C. $n = 57–96$ cells per condition for Day0 and $n = 11–48$ cells per condition for Days 1–7, total = 1448. **i** Box plot of number of UMIs detected corresponding to the cells in (**h**). **c, e–i** Data shown as median, interquartile range (IQR), and 1.5x IQR. **f–i** Data downsampled to 100,000 reads per cell (3456 cells were sequenced pre-downsampling) and red lines indicate medians for RRI. Data underlying relevant plots is provided in Source Data and $P$-values for statistical comparisons are available in Supplementary Data 1.

which was more prominent in the 4 °C condition, indicating that the interaction of cooling and SEQURNA was especially beneficial for delaying cell RNA degradation, in line with the previous observations of SS2 cDNA traces.

Thus, we conclude that substituting RRIs with the synthetic thermostable RNase inhibitor in the lysis buffer extends the workable time in between collecting and processing cells into scRNAseq libraries. While keeping lysates at above-freezing temperature is not recommended as a standard procedure as RNA degradation is inevitable over time, an extended timeframe has important logistical benefits when handling large sample batches, such as in sequencing facilities, which ultimately may improve information yield in single-cell research.

## Discussion

Here, we report that a synthetic thermostable RNase inhibitor (SEQURNA) can fully replace gold standard RRIs in multiple scRNAseq and bulk RNAseq applications without compromising library quality. This is likely to transform scRNAseq protocols going forward as the synthetic RNase inhibitor allows for various new workflow designs by its users. As part of this, we provide updated versions of the Smart-seq2, Smart-seq3, and Smart-seq3xpress protocols (Supplementary Notes 1–3).

While we have not applied all available scRNAseq platforms here, we expect synthetic RNase inhibitors to be compatible with any open platform method currently using RRIs. A key

consideration for adopting synthetic RNase inhibitors into existing and novel workflows is to identify the optimal inhibitor concentration that preserves RNA integrity while not affecting polymerase activity. Furthermore, it is worth noting that differences in buffer composition and volume changes between steps may affect the optimal concentration among protocols, as exemplified by Smart-seq2 and Smart-seq3. Because of its thermostability, the synthetic RNase inhibitor is added only in the cell lysis buffer (cell collection buffer) in these scRNAseq protocols and it retains RNase inhibition activity throughout heat cycles, which is in contrast to RRIs that need to be added first in the lysis buffer and then again in the RT mix following the cell-lysis and RNA-denaturation step by heating in the original protocols.

The synthetic RNase inhibitor has several advantages. Thermostability and robustness to various other harsh chemical and physical treatments enable more environmentally sustainable transportation and storage of the inhibitor at room temperature as well as novel and simplified workflows that are currently precluded by the less inert protein-based RRIs, which are thermosensitive and generally transported on dry ice. For example, stable lysis buffers and spotted plates can be pre-made, frozen, thawed, or kept at room temperature for more extended periods of time and need not be freshly prepared before cell collection, simplifying workflows and across-lab collaborations. These properties could also be beneficial in areas beyond scRNAseq, such as in vitro RNA synthesis and structural RNA applications.

An obvious strength of a chemically synthesized RNase inhibitor is that production- and storage-related batch effects can be kept at a minimum, while protein-based RRIs are more prone to degeneration and lot variation inherent to protein expression, purification, and storage over time. Notably, a chemically synthesized RNase inhibitor eliminates the risk of biological carryovers from the production process. Moreover, production-, storage-, and transportation-related costs can be reduced.

Thermostability and direct compatibility with Tn5 tagmentation further enables multiple use cases in single-cell multiomics techniques. Although we used a second-generation sequencing-based readout in the current study (short-read sequencing), we anticipate the utility of synthetic RNase inhibitors in third-generation long-read direct sequencing of RNA and cDNA.

We moreover envision that synthetic RNase inhibitors will be exceedingly beneficial in applications requiring large liquid volumes and high inhibitor consumption, which could facilitate in situ RNA sequencing in tissues or whole organisms.

In summary, the replacement of RRIs with synthetic thermostable RNase inhibitors represents a milestone in single-cell transcriptomics development.

## Methods

### Ethics statement
All animal experimental procedures were performed in accordance with Karolinska Institutet's guidelines and approved by the Swedish Board of Agriculture (permit 17956-2018, Jordbruksverket).

### Description of the synthetic RNase inhibitor
SEQURNA consists of a proprietary mix of synthetic molecules that interact with RNase, displacing amino acids in the active site. As such, we describe SEQURNA concentrations in terms of mass Units/µl (herein abbreviated U/µl) which is not neccessarily equivalent to activity Units of RRIs. Fresh tubes of RRI (Cat. 2313B TaKaRa) were used in all comparisons in this study.

### Animals and cells
C57BL/6 J (B6) mice were crossed with M. castaneous (CAST/EiJ) mice to produce F1 B6/CAST hybrid mice. Mice were housed in specific pathogen-free at Comparative Medicine Biomedicum (KM-B) according to Swedish national regulations for laboratory animal work food and water ad libitum, cage enrichment, and 12 h light and dark cycles. Primary fibroblasts were derived from adult CAST/EiJ × C57BL/6 J mice by skinning, mincing, and culturing tail explants in fibroblast medium (DMEM/10% FBS) in 5% CO2 and 37 °C. HEK293FT cells were cultured and expanded in standard medium (DMEM/10% FBS) in 5% CO2 and 37 °C, and 10 million cells were collected and resuspended in 0.5% BSA in PBS for cell sorting. Mouse liver and spleen were collected from purebred adult C57BL/6 J mice. For liver and spleen collection, 25 mL of warmed perfusion buffer (140 mM NaCl, 6.7 mM KCl, 9.6 mM HEPES, 6 mM NaOH) was slowly injected into the left ventricle after a small incision in the right ventricle. Both organs were collected on ice in perfusion buffer. The spleen was cut into pieces, mashed with a syringe plunger and filtered through a 70-µm cell strainer with the addition of 2% FBS in PBS. The liver was cut into pieces and treated with 1 mg/ml collagenase in perfusion buffer with 3 mM $CaCl_2$ and 1 mM $MgCl_2$ at 37 °C with shaking for 20 min. Liver cells were mashed through a 70-µm cell strainer with the syringe plunger, rinsing the strainer with 10% FBS in PBS. Both strained tissue types were pelleted, reconstituted in 1x RBC lysis buffer for 5 min at room temperature, and centrifuged at 500 g for 5 min to remove red blood cells. Spleen cells were washed with 0.5% BSA in PBS, reconstituted in 0.5% BSA in PBS, and filtered through a 40-µm cell strainer. Liver cells were washed 2x with PBS, incubated with 2 ml TrypLE express for 10 min at 37 °C with shaking, diluted to 20 ml with 10% FBS in PBS, pelleted, reconstituted in 0.5% BSA in PBS, and filtered

through a 40-µm cell strainer. The cell suspensions in 0.5% BSA in PBS were then subjected to single-cell sorting by FACS.

### Generation of Smart-seq2 and Smart-seq3 sequencing libraries
Bulk RNA was extracted from cultured mouse tail tip fibroblasts using TRIzol (Invitrogen). For all bulk RNAseq experiments 100 pg of RNA was used as input, if is not otherwise specified, and was added to 96-well plates containing either Smart-seq2 or Smart-seq3 standard lysis buffer or lysis buffer with varying concentrations of SEQURNA. In case of Supplementary Fig. 2i, we used 30 pg of TRIzol-extracted total RNA from mouse liver tissue as input. For Smart-seq2 and Smart-seq3 single-cell experiments, cultured HEK293FT, spleen, or liver cells in 0.5% BSA in PBS were sorted into 96 well plates containing lysis buffer using an SH800S Cell Sorter (Sony). Single-cell sorted plates were briefly centrifuged and kept at −80 °C until they were further processed. Plates containing lysis buffer and input material were then processed according to the Smart-seq2 and Smart-seq3 protocols[5,9], except without addition of recombinant inhibitor in the first-strand reaction mix in reactions where SEQURNA was used.

**Smart-seq2 cDNA library generation.** Smart-seq2 lysis buffer composition was 0.08% Triton X-100, 2.2 mM dNTP/each, 2.2 mM Smart-seq2 oligo-dT (5′-AAGCAGTGGTATCAACGCAGAGTACT30VN-3′), and 4 U recombinant inhibitor (Cat. 2313B, TaKaRa) or SEQURNA thermostable RNase inhibitor (Cat. SQ00201, SEQURNA) of the concentration specified for each condition; total buffer volume 4.5 µl. Cells were lysed and RNA denatured at 72 °C for 3 min in a Bioer Life ECO thermocycler and placed on ice prior to first-strand synthesis. The following reverse transcriptase reaction contained 1x Superscript II buffer, 5 mM DTT, 1 M betaine, 10 mM $MgCl_2$, 1 µM Smart-seq2 TSO (5′-AAGCAGTGGTATCAACGCAGAGTACATrGrG+G-3′), 100 U Superscript II, and 10 U recombinant inhibitor (TaKaRa) for samples containing biological inhibitor; total reaction volume 10 µl. RT thermocycles were 42 °C for 90 min, followed by 10 cycles of 42 °C for 2 min and 50 °C for 2 min. The following cDNA amplification reaction contained 1x KAPA HiFi HotStart Ready Mix and 80 nM ISPCR primers (5′-AAGCAGTGG-TATCAACGCAGAGT-3′); total reaction volume 25 µl. Thermocycles for Smart-seq2 cDNA amplification were 98 °C for 3 min, followed by either 18 (bulk), 20 (HEK), 21 (liver) or 22 (spleen) cycles of 98 °C for 20 sec, 67 °C for 15 sec, and 72 °C for 6 min, followed by a final incubation at 72 °C for 5 min.

**Smart-seq3 cDNA library generation.** Smart-seq3 lysis buffer composition was 0.1% Triton X-100, 10% PEG 8000, 1 mM dNTP/each, and 1 µM Smart-seq3 oligo-dT (5′-biotin-ACGAGCATCAGCAGCATACGA T30VN-3′), and 1.2 U recombinant inhibitor (Cat. 2313B, TaKaRa) or SEQURNA Thermostable RNase inhibitor (Cat. SQ00201, SEQURNA) of the concentration specified for each condition; total buffer volume 2 µl. Cells were lysed and RNA denatured at 72 °C for 10 min and placed on ice prior to first-strand synthesis. The following reverse transcriptase reaction contained 50 mM Tris-HCl (pH 8.3), 30 mM NaCl, 1 mM GTP, 8 mM DTT, 2.5 mM $MgCl_2$, 2 µM Smart-seq3 TSO (5′-biotin-AGAGACAGATTGCGCAATGNNNNNNNNNrGrGrG-3′), 8 U Maxima H-minus RT enzyme, and 2 U recombinant inhibitor (TaKaRa) for samples containing biological inhibitor; total reaction volume 4 µl. RT thermocycles were 42 °C for 90 min, followed by 10 cycles of 42 °C for 2 min and 50 °C for 2 min. The following cDNA amplification reaction contained 1x KAPA HiFi buffer, 300 nM dNTP/each, 500 nM MgCl2, 500 nM forward primer (5′-TCGTCGGCAGCGTCAGATGTGTATAAGA-GACAGATTGCGCAA*T*G-3′), 100 nM reverse primer (5′-ACGAGCAT-CAGCAGCATAC*G*A-3′), and 0.2 U KAPA polymerase in a total reaction volume of 10 µl. Thermocycles for Smart-seq3 cDNA amplification were 98 °C for 3 min, then either 20 (bulk) or 22 (single-cell) cycles of 98 °C for 20 sec, 65 °C for 30 sec, and 72 °C for 6 min, followed by a final incubation at 72 °C for 5 min.

Amplified cDNA was bead purified (AMPure XP, Beckman) at a ratio of 0.8:1 beads:cDNA (20 µl beads:25 µl cDNA for Smart-seq2 and 8 µl beads:10 µl cDNA for Smart-seq3) and inspected on a Bioanalyzer 2100 (High Sensitivity DNA kit, Agilent).

For Smart-seq2, the tagmentation reaction contained 2xTAPS-Mg buffer, 10% PEG 8000, 0.3 nM of a custom Tn5[11], and 1 ng of cDNA in 20 µl total, was incubated for 8 min at 55 °C, and Tn5 was stripped from the DNA with 3.5 µl of 0.2% SDS, incubated at room temperature for 5 min. To amplify Smart-seq2 libraries, a reaction mix containing the tagmented cDNA, 0.75 µl of 1 µM pooled dual-index primers, 1x Kapa HiFi PCR buffer, 300 µM dNTP/each, and 1 U Kapa HiFi polymerase in 50 µl total reaction was run in a thermocycler for 3 min at 72 °C, 95 °C 30 s, then 10 cycles of 95 °C for 10 s, 55 °C for 30 s, and 72 °C for 30 s, then 72 °C for 5 min.

For Smart-seq3, the tagmentation reaction contained 10 mM Tris-HCl pH 7.5, 5 mM MgCl2, 5% dimethylformamide, and 0.08 µl (0.12 µl for single cell) Tn5 (Nextera Amplicon Tagment Mix, Illumina) and 100 pg of cDNA in 2 µl total, was incubated for 10 min at 55 °C, and Tn5 was stripped from the DNA with 0.5 µl of 0.2% SDS (0.4 µl of 0.2% SDS for single-cell). To amplify Smart-seq3 libraries, a reaction mix containing the tagmented cDNA, 0.75 µl of 1 µM pooled dual index primers, 1x Phusion HiFi PCR buffer, 200 µM dNTP/each, and 0.1 U Phusion HiFi polymerase in 7 µl total was run on a thermocycler at 72 °C, 98 °C for 3 min, then 12 cycles of 98 °C for 10 s, 55 °C for 30 s, and 72 °C for 30 s, then 72 °C for 5 min.

Indexed libraries were bead purified (AMPure XP, Beckman) at a ratio of 0.6:1 beads:cDNA (30 µl beads:50 µl library for Smart-seq2 and 4.2 µl beads:7 µl library for Smart-seq3). All Smart-seq2 libraries were sequenced as 74 bp, single-end reads, and all Smart seq3 libraries were sequenced as 74 bp, paired-end reads.

### Bioanalyzer DNA yield calculations
All bioanalyzer traces were exported in raw format as a csv file. For all histograms, the average and standard error of the area-under-curve (DNA yield) was calculated for each sample. Percent primer dimer was calculated as (primer-dimer yield)/(total library yield). Number of replicas used for quantified cDNA and primer-dimer yield can be found in Supplementary Data 2.

### Read alignment and genes and UMIs detected
Read files were downsampled to 100k and aligned using STAR[12] against the Gencode GRm38 (mouse) or GRCh38 (human) genome assembly. STAR aligner determined the accuracy of mapping for each sample with percent uniquely mapped reads and percent unmapped reads. STAR was also used to calculate the rate of read mismatch to the genome and length of insertions and deletions within the read relative to the genome. MultiQC's output of STAR generated a summary of all STAR QC parameters for all samples. Data for both number of genes and UMIs detected was generated using the zUMIs pipeline[13]. Briefly, downsampled loom files that contained read coverage of exons only were processed in R for gene count (readcount.exon) and UMI count (umicount.exon). The number of genes expressed for each sample was quantified as sum of every gene with ≥1 read fragment. The number of UMIs detected for each sample was quantified as the sum of all UMI-containing fragments. Number of samples used for quality control analysis after downsampling (>100k reads) can be found in Supplementary Data 2.

### Base quality along read length
BBMap's tool mhist was used to plot the frequency that a read's base position matched or contained a substitution/insertion/deletion relative to the genome. Output files for each type were edited to and merged as a usable matrix for downstream processing in R.

### Genomic and gene body mapping
Sorted bam files generated by STAR were used in more in-depth mapping quantifications. The fraction of reads that mapped to exonic, intronic, or intergenic regions in the genome were obtained with Qualimap's RNA-seq QC. MultiQC's output of Qualimap generated a summary of all parameters for all samples. Gene body coverage plots were generated using computeMatrix and plotProfile via deepTools.

### Dimensionality reduction, cell cycle scoring, and clustering
For non-subsampled Smart-seq2 HEK data, cells with 3 MADs lower log10(genes detected) or log10(readcounts) than global median were excluded and readcounts were log-normalized then centered and scaled using Seurat[14] (v4.3; NormalizeData, ScaleData). Next, cells were scored for cell cycle using Seurat (CellCycleScoring, cc.genes.updated.2019) and PCA was performed using cell cycle-related genes used for cell cycle scores. For non-subsampled Smart-seq2 spleen and liver data, cells were filtered and normalized as described above, and 2000 variable features were selected using Seurat (FindVariableFeatures, selection.method = "vst"). PCA was performed using variable features and UMAP and nearest-neighbor detection was performed for the first 50 PCs using k = 30 nearest neighbors (RunUMAP, dims = 1:50, n.neighbors = 30; FindNeighbors, dims = 1:50, k.param = 30). Clusters were identified using Seurat (FindClusters, resolution = 0.3) and cluster identities were manually annotated based on gene expression. Number of samples passing quality control (<3 MAD below the median genes detected or readcounts) used for dimensionality reduction analyses can be found in Supplementary Data 2.

### PCR for primer dimer and non-specific band assays
The primer dimer assay was designed to produce a 164 bp amplicon from a Cas9-eGFP plasmid. The final PCR reaction mix contained 1x Hifi Kapa Mix, 1 µM of the fwd/rev primer pool, and 1 ng of plasmid. PCR reaction conditions were 95 °C for 3 min, then 25 cycles of 98 °C for 20 s, 65 °C for 15 s, and 72 °C for 30 s, followed by a final extension at 72 °C for 1 min. The non-specific band assays were designed to produce amplicons between 120 and 140 bp from Tuba1a in the human genome. The final PCR reaction mix for the non-specific band assay contained 1x Kapa Hifi Mix, 1 µM of the fwd/rev primer pool, and 20 ng of HEK293FT gDNA. PCR thermocycles were 95 °C for 3 min, then 25 cycles of 98 °C for 20 sec, 57 °C for 15 sec, and 72 °C for 30 sec, followed by a final extension at 72 °C for 1 min.

### Melt curve analysis
Purified eGFP amplicons (164 bp) were used as a dsDNA template. The reaction mixture contained 1x SYBR Mix, 500 ng of template, and varying concentrations of SEQURNA (n = 6 for each sample). The assay conditions were: 95 °C for 15 s to denature, 40 °C for 1 min to anneal, and a temperature gradient from 40 to 95 °C at a rate for 0.1° per second. Melt curve was performed with StepOne Plus Real-Time machine (Applied Biosystems). Results were exported to csv files using the Applied Biosystems Analysis Software.

### Smart-seq2 storage-time experiment
HEK293FT cells were FACS-sorted into Smart-seq2 lysis buffer (0.08% Triton X-100, 2.2 mM dNTP/each, and 2.2 mM Smart seq2 oligo-dT) containing 0, 6, 9, or 12 U/µl of SEQURNA in 96 well plates, and were subsequently stored at either 25 °C, 4 °C, or −80 °C. After 1, 4, 7, or 14 days, two replicas of each plate storage condition were processed for cDNA following the Smart-seq2 protocol (omitting the biological inhibitor in the reverse transcriptase step as described in a previous section) with 20 cycles of PCR amplification. An additional two plates were immediately processed into cDNA following the Smart-seq2 protocol after sorting as a control (day 0). After cDNA library amplification, samples were bead purified (AMPure XP, Beckman Coulter) and run on High Sensitivity DNA chips using a Bioanalyzer 2100 (Agilent) to assess cDNA quality.

## Smart-seq3xpress storage-time experiment and library preparation

Smart-seq3xpress libraries were performed as previously described[10] with some modifications. In brief, cells were sorted into 384 well plates each containing 3 μl Vapor-Lock (Qiagen) in all wells and eight different conditions of 0.3 μl lysis buffer consisting of 0.125 μM OligodT30VN (5′-Biotin-ACGAGCATCAGCAGCATACGAT30VN-3′; IDT) adjusted to RT volume, 0.5 mM dNTPs/each adjusted to RT volume, 0.1% Triton X-100, 5% PEG8000 adjusted to RT volume, and the indicated type and amount of RNase Inhibitor (no RNAse inhibitor, 0.003 μl RNase Inhibitor (40 U/μl, Cat. 2313B, TaKaRa), SEQURNA Thermostable RNase inhibitor (Cat. SQ00201, SEQURNA); 0.06 U/μl, 0.15 U/μl, 0.3 U/μl, 0.6 U/μl, 1.5 U/μl, 3.0 U/μl). After cell sorting plates were briefly centrifuged before put to −80 °C immediately after sorting (day 0). To test the effect of temperature and time on single cell RNA stability for each of the eight lysis conditions, 384 well plates containing sorted cells in lysis buffer were left at either room temperature (25 °C) for 1, 4, or 7 days or in fridge (4 °C) for 1, 4, 7, or 14 days, before commencing library preparation. To serve as control (day 0), two plates were extracted immediately from −80 °C immediately before library preparation.

Before RT, plates were denatured at 72 °C for 10 min followed by addition of 0.1 μl of RT mix; 25 mM Tris-HCL pH 8.4 (Fischer Scientific), 30 mM NaCl (Ambion), 1 mM GTP (Thermo Fisher Scientific), 2.5 mM MgCl$_2$ (Ambion), 8 mM DTT (Thermo Fisher Scientific), 0.75uM Template Switching Oligo (TSO) (5′-Biotin-AGAGACAGATTGCGCAATG NNNNNNNNNWWrGrGrG-3′; IDT), 0.25 U/ μl RRI (Cat. 2313B, TaKaRa), and 2 U/μl of Maxima H Minus reverse transcriptase (Cat. EP0752, Thermo Fisher Scientific). RRI was excluded from the RT mix (replaced with water) in case of SEQURNA samples. Plates were quickly centrifuged after dispensing to ensure merge of lysis and RT volumes underneath the Vapor-lock overlay and incubated at 42 °C for 90 min, followed by ten cycles of 50 °C for 2 min and 42 °C for 2 min. After RT, 0.6 μl PCR mix was dispensed to each well containing the following: 1× SeqAmp PCR buffer (Takara Bio), 0.025 U μl−1 of SeqAmp polymerase (Takara Bio) and 0.5 μM Smartseq3 forward (5′-TCGTCGGCAGCGT-CAGATGTGTATAAGAGACAGATTGCGCAATG-3′; IDT) and reverse primer (5′-ACGAGCATCAGCAGCATACGA-3′; IDT). Plates were quickly spun down before being incubated as follows: 1 min at 95 °C for initial denaturation, 12 cycles of 10 s at 98 °C, 30 s at 65 °C and 4 min at 68 °C. Final elongation was performed for 10 min at 72 °C.

After PCR, pre-amplified libraries were diluted with 9 μl H2O, before transferring 1 μl of diluted cDNA from each well into a new 384 well plate. Tagmentation was performed by adding 1 μl of tagmentation mix; 1x tagmentation buffer (10 mM Tris pH 7.5, 5 mM MgCl2, 5% DMF), 0.003 μl Tagmentation DNA Enzyme 1 (TDE1; Illumina DNA sample preparation kit) to the 1 μl of diluted cDNA per well. At this step the tagmentation mix contains SEQURNA RNase inhibitor at a concentration of 0.015x what was in the lysis buffer. Plates were incubated for 10 min at 55 °C before the reaction was stopped by the addition of 0.5 μl 0.2% SDS to each well. Index PCR was carried out after the addition of 3.5 μl custom Nextera Index primers (0.5 μM) by dispensing 2 μl of PCR mix; 1× Phusion Buffer (Thermo Fisher Scientific), 0.01 U μl−1 of Phusion DNA polymerase (Thermo Fisher Scientific), 0.025% Tween-20, 0.2 mM dNTP each. PCR was performed out at 3 min at 72 °C; 30 s at 95 °C; 12 cycles of (10 s at 95 °C; 30 s at 55 °C; 1 min at 72 °C); and 5 min at 72 °C. Each indexed library plate was pooled by spinning out gently using a 300-ml robotic reservoir (Nalgene) fitted with a custom scaffold by pulsing the centrifuge to <200 g. The pooled libraries were afterward purified with custom carboxylated magnetic beads in 22% PEG solution at a ratio of 1 sample to 0.7 beads.

## Sequencing of Smart-seq3xpress libraries

Smart-seq3xpress libraries were sequenced on a MGI DNBSEQ G400RS platform. Prior to sequencing on MGI platform ready circular ssDNA libraries were generated using the MGIEasy Universal Library Conversion Kit (MGI). Adapter conversion PCR was carried out on 50 ng of final pooled library for 5 cycles, following circularization of 1pmol dsDNA according to manufacturer's protocol. DNA nanoballs (DNBs) were created from 80 fmol of circular ssDNA library pools using a custom rolling-circle amplification primer (5′-TCGCCGTATCATT-CAAGCAGAAGACG-3′, IDT). DNBs were sequenced 100 bases paired end (PE100) using custom sequencing primers (Read 1: 5′-TCGT CGGCAGCGTCAGATGTGTATAAGAGACAG-3′; MDA: 5′-CGTATGCCG TCTTCTGCTTGAATGATACGGCGAC-3′, Read 2: 5′-GTCTCGTGG GCTCGGAGATGTGTATAAGAGACAG-3′; i7 index: 5′-CCGTATCATT-CAAGCAGAAGACGGCATACGAGAT-3′; i5 index: 5′-CTGTCTCTTATA-CACATCTGACGCTGCCGACGA-3′).

## Data preprocessing of Smart-seq3xpress libraries

Raw FASTQ files were processed with zUMIs[13] 2.9.7 pipeline. UMI-containing reads were identified by the (ATTGCGCAATG) pattern allowing up to two mismatches and reads were filtered for low quality UMIs (3 bases <phred 20) and index barcodes (4 bases <phred 20), before mapped to the human genome (hg38) using STAR5 version 2.7.3. Read counts, detected genes, and umicounts were calculated using Ensembl gene annotations (GRCh38.95). Down-sampled data (100k reads or 100k UMI-containing reads) was generated via zUMIs pre-processing. Number of samples used for quality control analysis after downsampling ( >100k reads) can be found in Supplementary Data 2.

## Inhibitor-spiked tagmentation

Bulk HEK RNA (100 pg) was used to generate amplified cDNA with the standard Smart-seq2 protocol (18 cycles, full 96-well plate) replacing RRI in the lysis buffer with SEQURNA (1.2 U/μl). All wells were pooled and bead purified (AMPure XP, Beckman) at a ratio of 0.8:1 beads:cDNA. Purified, amplified cDNA was diluted to a 1 ng/μl stock to use for tagmentation.

The tagmentation reaction contained 2xTAPS-Mg buffer, 10% PEG 8000, 0.3 nM Tn5, and 2 ng of cDNA. SEQURNA was added to the reactions at final concentrations of 0.0025, 0.01, 0.025, 0.1, 0.25, 1, or 2.5 U/μl (or no inhibitor) in 20 μl total. The reaction was incubated for 8 min at 55 °C, and Tn5 was stripped from the DNA with 3.5 μl of 0.2% SDS, incubated at room temperature for 5 min. Tagmented libraries were amplified in a reaction mix containing 1 μl of pooled custom dual-index primers, 1x Kapa HiFi PCR buffer, 300uM dNTP/each, and 1 U Kapa HiFi polymerase in 50 μl total, and was run in a thermocycler for 3 min at 72 °C, 95 °C 30 s, then 10 cycles of 95 °C for 10 s, 55 °C for 30 s, and 72 °C for 30 s, then 72 °C for 5 min. Indexed libraries were bead purified (AMPure XP, Beckman) at a ratio of 1:1 beads:library (50 μl beads:50 μl library). Library concentration was quantified with the QuantiFluor ONE dsDNA System using the Varioskan LUX (Thermo Scientific) microplate reader. Library quality was visualized on a Bioanalyzer 2100 (High Sensitivity DNA kit, Agilent).

## Data visualization

All data were plotted using ggplot2 or ComplexHeatmap in R.

## Statistics and reproducibility

Statistics were performed as two-tailed Mann–Whitney U-tests available in Supplementary Data 1. Sample sizes available in Supplementary Data 2, no statistical method was used to predetermine sample size. Low quality single-cell libraries were excluded from analysis based on the criteria stated above. The experiments were not randomized, and the Investigators were not blinded to allocation during experiments and outcome assessment.

## Reporting summary

Further information on research design is available in the Nature Portfolio Reporting Summary linked to this article.

## Data availability

The raw and processed sequencing data generated in this study have been deposited in the ArrayExpress database under accession codes E-MTAB-13873 (single-cell RNA-seq data) and E-MTAB-13899 (bulk RNA-seq). All other data generated in this study are provided in the Source Data file. Source data are provided with this paper.

## Code availability

Computational code generated in this study is available at github.com/reiniuslab/SEQURNA (Permanent release https://doi.org/10.5281/zenodo.13380268).

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

## Acknowledgements

A.L. is supported by the Swedish Society for Medical Research (PD20-0217). B.R. is supported by the Swedish Society for Medical Research (CG-22-0260-H-02), the Knut & Allice Wallenberg Foundation (2021.0142 and 2022.0146), the Swedish Research Council (2022-01620), and KI SFO StratRegen 2021. We thank members from Reinius' lab for input on the paper.

## Author contributions

J.C.N., A.L., and B.R. conceived the study. J.C.N. performed experiments. M.H.J. and R.S. performed Smart-seq3xpress experiments. J.C.N., A.L., and B.R. analyzed the data, prepared figures, and wrote the paper. B.R. and R.S. provided resources. B.R. supervised the work. All authors read and edited the manuscript.

## Funding

## Competing interests

J.C.N., A.L., and B.R. have filed patent applications on synthetic RNase inhibitors in scRNAseq and other applications and are co-founders of SEQURNA AB, making available the thermostable RNase inhibitor mix herein described as a quality-controlled kit. The composition of the inhibitor mix is undisclosed. J.C.N., A.L., M.H.J., R.S., and B.R. are shareholders of SEQURNA AB.
