## [Peer Review File · Nature Communications]

REVIEWER COMMENTS

Reviewer #1 (Remarks to the Author):

Next-generation sequencing-based genomics, prominently single-cell RNA-seq, has revolutionized our understanding of biological processes. A comprehensive refinements, both technical and analytical, has been necessary for the maturation and wide applicability of these experimental approaches. Due to the extremely low RNA material in a single cell and the high RNase content present in many tissues, data quality is highly dependent on RNA integrity preservation. RNA degradation is addressed by the addition of RNase inhibitors, which invariably are commercial recombinant proteins (RRI for recombinant RNase inhibitors). As proteins are sensitive to denaturing factors, their activity cannot be warranted, compromising both data quality and reproducibility. In this manuscript, Noble and colleagues address this question by developing an RNase inhibitor compound (SEQURNA). This compound is not a protein and thus is more thermostable than RRI, improving sample quality and portability.

The authors comprehensibly characterize SEQURNA, providing transparent information for its applicability and for end users. For example, they demonstrate that RNase inhibitor concentration should be optimized per application (exemplified here by SS2 vs SS3). SEQURNA presents a novel way of overcoming the limitations imposed by using protein-based Rnase inhibitors and is an important addition to the single-cell toolkit for rendering high RNase-content samples accesible, and facilitate sample transfer in large international collaborative research projects. Therefore, I support its publication provided a few issues listed below are managed.

Major points:

1. The manuscript should mention that this inhibitor is currently not applied in all platforms. It is not applicable in closed commercial single-cell platforms such as 10X, in which the lysis and RT mix composition cannot be modified by the end user.
2. Line 54 spiked mouse RNA – Unclear what “spiked” stands for. Please clarify (Methods section or here).
3. In supplementary figure 2, in which performance is tested under stress conditions, the comparison to appreciate the improvement brought about by SEQURNA is missing. How does the trace of a SS2 library in which the RNase inhibitor was RRI look like?

Minor points:

1. Line 19: Miniscule should be corrected to formal minuscule
2. End of Line 46: concentration instead of concentrations
3. Line 60 To improve readability, the authors should clarify that primer-dimer is not tested with SS2 for figures 1d and 1e, but with regular PCR.

Reviewer #2 (Remarks to the Author):

Noble et al introduce a synthetic thermostable RNase inhibitor and perform systematic comparisons of the performance of this inhibitor to conventional recombinant RNase inhibitors for bulk and plate-based single-cell RNA-seq applications. Overall, this is an interesting paper and both the experimental and computational methods are technically sound. However, I found it difficult to appreciate how this represents a “substantial advancement in the field of single-cell transcriptomics” as claimed by the authors, at least with the results presented here.

Major comments:

1) Figures 1-3 compare the performance of different concentrations of their synthetic RNase inhibitor to that of a conventional recombinant RNase inhibitor in three applications – bulk RNA-seq, single-cell RNA-seq with Smart-seq2, and single-cell RNA-seq with Smart-seq3. They show standard performance characteristics such as the number of genes detected per cell or per sample, cDNA length distributions, and gene body coverage distributions. Overall, the synthetic RNase inhibitor exhibits comparable performance to that of the conventional recombinant inhibitor. So while it is nice that an alternative inhibitor exists, there is no fundamentally new capability demonstrated by these experiments.

2) Figure 4 highlights a comparison in which a difference in performance between the synthetic RNase inhibitor and the conventional recombinant RNase inhibitor can be appreciated. Here, cells are sorted into lysis buffers containing either of the two inhibitors in multi-well plates and stored under various conditions, including -80C (the zero day time point), 4C, and 25C, and for different amounts of time. While the two inhibitors perform comparably at -80C, there is an increasing difference in performance at longer times (i.e. 1-2 weeks) at 4C and 25C. While this is interesting, it is standard practice to solve this problem by simply storing the samples at -80. For example, why would one use this inhibitor and store lysates at 25C for a week when one can simply place the samples at -80C with a conventional recombinant inhibitor and get better performance, as implied by Fig. 4f-g?

Reviewer #3 (Remarks to the Author):

In this manuscript, Dr. Noble and colleagues delve into the utility of a synthetic thermostable RNase inhibitor for single-cell library generation. They present compelling evidence that this inhibitor matches and often outperforms the protein-based RNase inhibitors (RRIs). These RRIs, while effective in minimizing RNA degradation during single-cell library preparation, come with significant drawbacks. They are expensive, prone to degradation, and can introduce batch variability. Moreover, their thermal sensitivity limits the temperature ranges used in the procedures. Although attempted, alternative methods such as guanidinium salts, beta-mercaptoethanol, or diethyl pyrocarbonate have failed to match the performance of RRIs.

The synthetic RNase inhibitor, SEQURNA, presented in this manuscript, is thermostable. The authors demonstrate the inhibitor's utility for mini-bulk RNA seq library generation, and the results are striking. SEQURNA's quality matches or exceeds standard RRIs in SS2 libraries. The authors then take it a step further, evaluating the performance of SEQURNA in single-cell SS2 and SS3, and the findings are consistent with the previous results. SEQURNA's quality matches or exceeds that of standard RRIs. Finally, they tested the utility of SEQURNA for long-term storage of RNA. Once again, SEQURNA significantly slows RNA degradation during long-term storage compared to standard RRIs.

This new synthetic RNase inhibitor can be useful to the RNA sequencing community. I have just a few comments to improve the manuscript:

1. Statistics should be used across all figures to compare RRI's performance directly to that of the new inhibitor.
2. Figure 1c shows that primer dimer decreases, but in Figure 1b, there is little cDNA product at these concentrations. Specifically, from Figure 1b, the optimal inhibitor concentration is 2U/ul, but at 2U/ul, the primer dimer looks to be the same or slightly higher than the RRI. The text is a bit misleading because this is described as outperforming RRIs. Writing the exact numerical values into the text would be helpful, and the authors should soften the claim that this is a novel property since RRIs also show a similar level of primer dimer.
3. The usable range for SEQURNA for SS3 (fig 3b-0.10-1U/ul) is lower than for SS2 (~2-3U/ul). It would be helpful for the authors to discuss why in the discussion.
4. For Figure 4b, please discuss why the low concentration of SEQURNA improves the library yield. In other words, why might a high concentration hurt it?

POINT-BY-POINT RESPONSES TO THE REVIEWER'S COMMENTS

Noble et al 2024. Introducing synthetic thermostable RNase inhibitors to single-cell RNA-seq

Reviewer #1 (Remarks to the Author):

Next-generation sequencing-based genomics, prominently single-cell RNA-seq, has revolutionized our understanding of biological processes. A comprehensive refinements, both technical and analytical, has been necessary for the maturation and wide applicability of these experimental approaches. Due to the extremely low RNA material in a single cell and the high RNase content present in many tissues, data quality is highly dependent on RNA integrity preservation. RNA degradation is addressed by the addition of RNase inhibitors, which invariably are commercial recombinant proteins (RRI for recombinant RNase inhibitors). As proteins are sensitive to denaturing factors, their activity cannot be warranted, compromising both data quality and reproducibility. In this manuscript, Noble and colleagues address this question by developing an RNase inhibitor compound (SEQURNA). This compound is not a protein and thus is more thermostable than RRI, improving sample quality and portability.

The authors comprehensively characterize SEQURNA, providing transparent information for its applicability and for end users. For example, they demonstrate that RNase inhibitor concentration should be optimized per application (exemplified here by SS2 vs SS3). SEQURNA presents a novel way of overcoming the limitations imposed by using protein-based Rnase inhibitors and is an important addition to the single-cell toolkit for rendering high RNase-content samples accesible, and facilitate sample transfer in large international collaborative research projects. Therefore, I support its publication provided a few issues listed below are managed.

We thank Reviewer #1 for finding our work a novel and important addition to the field as well as the information provided both comprehensive and transparent. Thanks also for a clear assessment and for your suggestions on how to further improve the manuscript. We have addressed each of your questions, including performing additional experiments.

Major points:

1. The manuscript should mention that this inhibitor is currently not applied in all platforms. It is not applicable in closed commercial single-cell platforms such as 10X, in which the lysis and RT mix composition cannot be modified by the end user.

Correct. We have added the following text to clarify that we do not include data for all platforms and that closed platform methods are exempt:

“While we have not applied all available scRNAseq platforms here, we expect synthetic RNase inhibitors to be compatible with any open platform method currently using RRIs.”

2. Line 54 spiked mouse RNA – Unclear what “spiked” stands for. Please clarify (Methods section or here).

With “spiked” we meant fixed input amount of RNA, but is redundant to state as we also state 100 pg as the input. We thus agree with the reviewer and have clarified the text to:

“100 pg purified mouse RNA”

3. In supplementary figure 2, in which performance is tested under stress conditions, the comparison to appreciate the improvement brought about by SEQURNA is missing. How does the trace of a SS2 library in which the RNase inhibitor was RRI look like?

Great suggestion. We have now added such experimental data in the new panel **Supplementary Fig. 2i**, in which we show resulting SS2 cDNA libraries using SEQURNA and RRI stocks that had been heated at different temperatures and time before usage, demonstrating retained RNase inhibition with SEQURNA. Notably, in these experiments we used total RNA extracted from liver, a tissue known to be enriched in RNases, at single-cell input amounts (30 pg) demonstrating solid performance of SEQURNA. We did not test effects on pH changes and salt concentration changes in the RRI condition, as the composition of the commercial RRI buffer is undisclosed and would furthermore be different in various brands of RRIs, making that exercise circumstantial and difficult. Thanks for suggesting this experimental addition.

Minor points:

1. Line 19: Miniscule should be corrected to formal minuscule

Done. Thank you for highlighting this.

2. End of Line 46: concentration instead of concentrations

Done.

3. Line 60 To improve readability, the authors should clarify that primer-dimer is not tested with SS2 for figures 1d and 1e, but with regular PCR.

Thank you for notifying that this was unclearly formulated. We have now clarified this point:

“To better explore how SEQURNA affects PCR specificity, we performed a previously established PCR-based dimer assay using primers intentionally designed to self-hybridize⁶ and an eGFP plasmid as template (**Methods**).”

We thank Reviewer #1 for your helpful comments, and experimental suggestions. We believe that we have addressed all comments thoroughly. Your input has helped clarify several points and strengthen the manuscript overall. We hope that you now find it fit for publication.

Reviewer #2 (Remarks to the Author):

Noble et al introduce a synthetic thermostable RNase inhibitor and perform systematic comparisons of the performance of this inhibitor to conventional recombinant RNase inhibitors for bulk and plate-based single-cell RNA-seq applications. Overall, this is an interesting paper and both the experimental and computational methods are technically sound. However, I found it difficult to appreciate how this represents a “substantial advancement in the field of single-cell transcriptomics” as claimed by the authors, at least with the results presented here.

We thank Reviewer #2 for finding our paper interesting and supported with sound experimental and computational methodology. We hope that our responses as well as the comments from the other reviewers will help clarify the advancements our manuscript makes in the field.

Major comments:

1) Figures 1-3 compare the performance of different concentrations of their synthetic RNase inhibitor to that of a conventional recombinant RNase inhibitor in three applications – bulk RNA-seq, single-cell RNA-seq with Smart-seq2, and single-cell RNA-seq with Smart-seq3. They show standard performance characteristics such as the number of genes detected per cell or per sample, cDNA length distributions, and gene body coverage distributions. Overall, the synthetic RNase inhibitor exhibits comparable performance to that of the conventional recombinant inhibitor. So while it is nice that an alternative inhibitor exists, there is no fundamentally new capability demonstrated by these experiments.

We respectfully disagree with the Reviewer on this point as our paper provides the first alternative to recombinant RNase inhibitors that is 1) greatly thermostable, 2) works under native, non-destructive, conditions, 3) shows comparable or even improved performance in several aspects of scRNAseq workflows, 4) enables development of novel experimental procedures and protocols including heat cycles during which RNase inhibition is retained, and 5) will reduce the need of dry-ice shipments for RNase inhibitors simplifying logistics and improving the sustainability aspect of RNA work due to markedly reduced energy consumption. Due to its thermostability, the synthetic RNase inhibitor is only added in the lysis buffer and remain active throughout the lysis and denaturation step by heating as well as the following RT step. Conversely, recombinant RNase inhibitors require DTT for functionality and must be added first in cell lysis and then once again in the RT mix (as the RRI loses its activity upon heating in the lysis and denaturation step),

resulting in a larger fraction of the cost per single cell library. Additionally, the thermostable RNase inhibitor now allows for pre-made lysis buffer mixes including the inhibitor, which can be stored and shipped at ambient condition. Previous lysis buffer mixes relying on RRIs, on the other hand, need to either be freshly prepared before cell collection or kept in frozen condition spotted on the plates. Furthermore, the synthetic inhibitor extends the workable timeframe after cell collection, as we demonstrate experimentally. These aspects do not only simplify sample processing, but the thermostability will allow for various new workflow designs by the users of the synthetic RNase inhibitor ahead. We hope that the reviewer agrees that the added benefits of the synthetic RNase inhibitor results in a clear move forward in scRNAseq methodology.

2) Figure 4 highlights a comparison in which a difference in performance between the synthetic RNase inhibitor and the conventional recombinant RNAseq inhibitor can be appreciated. Here, cells are sorted into lysis buffers containing either of the two inhibitors in multi-well plates and stored under various conditions, including -80C (the zero day time point), 4C, and 25C, and for different amounts of time. While the two inhibitors perform comparably at -80C, there is an increasing difference in performance at longer times (i.e. 1-2 weeks) at 4C and 25C. While this is interesting, it is standard practice to solve this problem by simply storing the samples at -80. For example, why would one use this inhibitor and store lysates at 25C for a week when one can simply place the samples at -80C with a conventional recombinant inhibitor and get better performance, as implied by Fig. 4f-g?

Having the option to store lysates at 4°C or room temperature for some time can clearly be a benefit in large-scale and automated production settings and during sample transportation where samples are at risk of being thawed for a prolonged period. We are however in agreement with the reviewer that this is not the recommended procedure and have added the following text to clarify this:

“While storing lysates at increased temperature is not recommended as a standard procedure, this has important logistical benefits when handling large sample batches, such as in sequencing facility, which ultimately may improve information yield in single-cell research.”

We thank Reviewer #2 for your comments that have helped clarify and strengthen several key points in our manuscript and we hope that you now find it suitable for public release.

Reviewer #3 (Remarks to the Author):

In this manuscript, Dr. Noble and colleagues delve into the utility of a synthetic thermostable RNase inhibitor for single-cell library generation. They present compelling evidence that this inhibitor matches and often outperforms the protein-based RNase inhibitors (RRIs). These RRIs, while effective in minimizing RNA

degradation during single-cell library preparation, come with significant drawbacks. They are expensive, prone to degradation, and can introduce batch variability. Moreover, their thermal sensitivity limits the temperature ranges used in the procedures. Although attempted, alternative methods such as guanidinium salts, beta-mercaptoethanol, or diethyl pyrocarbonate have failed to match the performance of RRIs.

The synthetic RNase inhibitor, SEQURNA, presented in this manuscript, is thermostable. The authors demonstrate the inhibitor's utility for mini-bulk RNA seq library generation, and the results are striking. SEQURNA's quality matches or exceeds standard RRIs in SS2 libraries. The authors then take it a step further, evaluating the performance of SEQURNA in single-cell SS2 and SS3, and the findings are consistent with the previous results. SEQURNA's quality matches or exceeds that of standard RRIs. Finally, they tested the utility of SEQURNA for long-term storage of RNA. Once again, SEQURNA significantly slows RNA degradation during long-term storage compared to standard RRIs.

This new synthetic RNase inhibitor can be useful to the RNA sequencing community. I have just a few comments to improve the manuscript:

We thank Reviewer #3 for the enthusiastic comments and for underscoring the usefulness of the synthetic thermostable RNase inhibitor to the field. We have addressed each of your suggestions on how to improve the paper.

1. Statistics should be used across all figures to compare RRI's performance directly to that of the new inhibitor.

We have now added Mann Whitney U-test statistics comparing SEQURNA concentrations to RRI for all main figures into the new item **Supplementary Data 2**, so that these data are now available with the paper. However, we do not think that adding all P-values directly in the figures will be very helpful to the readers, as it is the ranges and trends that are more relevant in the analyses rather the statistical significance of any specific mean- or median-value comparisons. Notably, even irrelevantly small changes in median values are bound to come out significant when the number of samples is high. To accommodate the reviewer's assessment, we have provided graphics of the relevant main-figure panels with P-values on the following page (**Reviewer Figure 1**). Due to figure space limitations, we abbreviate statistical significance to $P < 0.05$ in the reviewer figure, but we provide exact P-values in the new **Supplementary Data 2**. We hope that the reviewer agrees that inclusion of new **Supplementary Data 2** is an appropriate solution to the point raised.

Reviewer Figure 1

2. Figure 1c shows that primer dimer decreases, but in Figure 1b, there is little cDNA product at these concentrations. Specifically, from Figure 1b, the optimal inhibitor concentration is 2U/ul, but at 2U/ul, the primer dimer looks to be the same or slightly higher than the RRI. The text is a bit misleading because this is described as outperforming RRIs. Writing the exact numerical values into the text would be helpful, and the authors should soften the claim that this is a novel property since RRIs also show a similar level of primer dimer.

We agree that this comment is a fair point and have therefore toned down the claims related to primer-dimers in scRNAseq cDNA libraries in the main text. Nonetheless, it is still clear that SEQURNA has an effect on dimer-dimer formation, which is evident in the PCR-based dimer assay. The mentioned section now reads:

“Intriguingly, we noticed that SEQURNA also slightly reduced the fraction of primer-dimer content in the cDNA libraries at high concentrations (**Fig. 1a, c**). To better explore how SEQURNA affects PCR specificity, we performed a previously established PCR-based dimer assay...”

Thanks for helping us improve the readability of this section.

3. The usable range for SEQURNA for SS3 (fig 3b–0.10-1U/ul) is lower than for SS2 (~2-3U/ul). It would be helpful for the authors to discuss why in the discussion.

Indeed, this is a key factor to consider when using this inhibitor in new assays. We have added the following text in the Discussion, highlighting these differences:

“While we have not applied all available scRNAseq platforms here, we expect synthetic RNase inhibitors to be compatible with any open platform method currently using RRIs. A key consideration for adopting synthetic RNase inhibitors into existing and novel workflows is to identify the optimal inhibitor concentration that preserves RNA integrity while not affecting polymerase activity. Furthermore, it is worth noting that differences in buffer composition and volume may affect the optimal concentration between protocols, as exemplified by Smart-seq2 and Smart-seq3. Because of its thermostability, the synthetic RNase inhibitor is added only in the cell lysis buffer (cell collection buffer) in these scRNAseq protocols and it retains RNase inhibition activity throughout heat cycles, which is in contrast to RRIs that need to be added first in the lysis buffer and then again in the RT mix following the lysis- and RNA-denaturation step by heating in the original protocols.”

4. For Figure 4b, please discuss why the low concentration of SEQURNA improves the library yield. In other words, why might a high concentration hurt it?

We do not yet fully understand this observed effect of SEQURNA on Tn5 and to identify the mechanism would require a major effort that we consider being well beyond the focus of our current paper (i.e. demonstrating that synthetic RNase inhibitors are a viable option to RRI in scRNAseq). Although we agree with the reviewer that the question is interesting, we opt to not speculate about a mechanism before having a better understanding of it. However, we have added wordings in the main text to state that the exact mechanism remains to be determined:

“While the exact mechanism behind this interaction remains to be determined, this effect might be beneficial in single-cell multi-omics experiments utilizing Tn5.”

We thank Reviewer #3 for your helpful comments that have helped improve our manuscript and we hope that you now find it fit for release.

REVIEWERS' COMMENTS

Reviewer #1 (Remarks to the Author):

I wish to congratulate the authors for this important study. I approve the manuscript in its current form.

Reviewer #2 (Remarks to the Author):

I don't have any further technical concerns with the manuscript, and the authors have appropriately edited the paper in light of the reviewer comments. I still feel that the impact of the advance reported here is somewhat over-stated by the authors, and I doubt that sequencing facilities are going to store cell lysates at room temperature.

Reviewer #3 (Remarks to the Author):

The authors have addressed everything, I have no further questions!

REVIEWERS' COMMENTS

Reviewer #1 (Remarks to the Author):

I wish to congratulate the authors for this important study. I approve the manuscript in its current form.

We thank the reviewer for all the comments provided on our previous version, helping us to improve the paper.

Reviewer #2 (Remarks to the Author):

I don't have any further technical concerns with the manuscript, and the authors have appropriately edited the paper in light of the reviewer comments. I still feel that the impact of the advance reported here is somewhat over-stated by the authors, and I doubt that sequencing facilities are going to store cell lysates at room temperature.

We sincerely thank the reviewer for the constructive feedback that has helped us refine the paper. We completely agree that it is not a recommended procedure to store cell lysates at room temperature, and we believe that the text emphasizes this.

Reviewer #3 (Remarks to the Author):

The authors have addressed everything, I have no further questions!

We appreciate the reviewer's detailed comments and suggestions, which have contributed to improving the manuscript.